# FASP: FAST AND ACCURATE STRUCTURED PRUNING OF LARGE LANGUAGE MODELS

## ABSTRACT

The rapid increase in the size of large language models (LLMs) has significantly escalated their computational and memory demands, posing challenges for efficient deployment, especially on resource-constrained devices. Structured pruning has emerged as an effective model compression method that can reduce these demands while preserving performance. In this paper, we introduce FASP (Fast and Accurate Structured Pruning), a novel structured pruning framework for LLMs that emphasizes both speed and accuracy. FASP employs a distinctive pruning structure that interlinks sequential layers, allowing for the removal of columns in one layer while simultaneously eliminating corresponding rows in the preceding layer without incurring additional performance loss. The pruning metric, inspired by Wanda, is computationally efficient and effectively selects components to prune. Additionally, we propose a restoration mechanism that enhances model fidelity by adjusting the remaining weights post-pruning. We evaluate FASP on the OPT and LLaMA model families, demonstrating superior performance in terms of perplexity and accuracy on downstream tasks compared to state-of-the-art methods. Our approach achieves significant speed-ups, pruning models such as OPT-125M in 17 seconds and LLaMA-30B in 20 minutes on a single NVIDIA RTX 4090 GPU, making it a highly practical solution for optimizing LLMs.

## 1 INTRODUCTION

Large language models (LLMs) have profoundly transformed various aspects of daily life in recent times, showcasing remarkable capabilities across diverse applications, including machine translation, conversational agents, text generation, as well as image and video synthesis (OpenAI, 2023; Meta AI, 2023; Gemini Team et al., 2023; Arefeen et al., 2024; Li et al., 2024). However, the increasing size of these models imposes substantial demands on computational and GPU memory resources, creating significant barriers to the deployment of advanced LLMs. This challenge becomes even more pronounced when considering the deployment of LLM technologies on mobile devices, such as smartphones, particularly in light of the rising trend driven by the release of advanced AI capabilities like Apple intelligence (Apple Inc, 2024). In light of these challenges, the compression of LLMs has become a critical requirement to mitigate the resource demands associated with their deployment, thereby enhancing their accessibility and operational efficiency across diverse platforms.

Pruning techniques represent one of the most prevalent methods for compressing LLMs (Frantar & Alistarh, 2023; Sun et al., 2023; Ma et al., 2023a; Shen et al., 2024; Fang et al., 2024), effectively reducing model size and computational demands while maintaining performance integrity. Among the various pruning strategies, structured pruning stands out for its systematic approach of removing entire components, such as neurons or channels, which directly enhances computational efficiency and is compatible across diverse hardware platforms. In contrast, unstructured pruning targets individual weights, necessitating support for sparse data structures and corresponding computational methods. Similarly, semi-structured pruning often relies on specific hardware architectures, such as the 2:4 semi-structure utilized in NVIDIA's Ampere architecture (Mishra et al., 2021). Additionally, the inference acceleration achieved through semi-structured pruning is less effective than that obtained with structured pruning at the same level of sparsity (Ashkboos et al., 2024). Therefore, this work will concentrate on structured pruning.

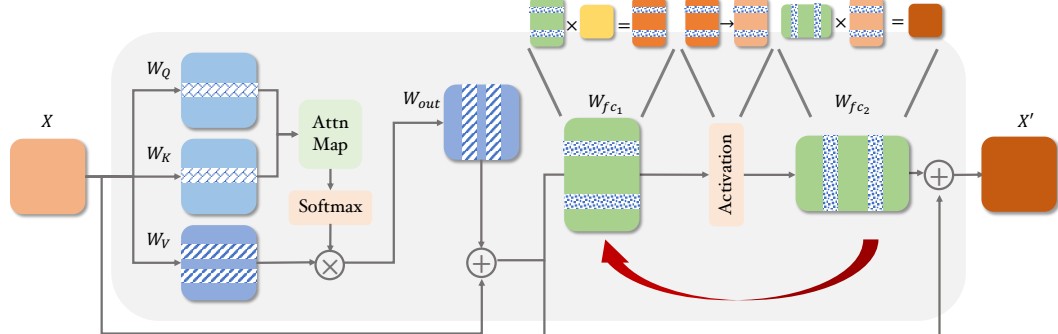

Figure 1: Illustration of the proposed pruning structure on the OPT model. In this approach, columns of $W_{fc_2}$ are removed along with the corresponding rows of $W_{fc_1}$ without impacting performance, thanks to the inherited position mapping in matrix multiplication. The same principle applies to columns of $W_{\text{out}}$ and rows of $W_V$, as well as the rows of $W_Q$ and $W_K$.

Several existing structured pruning methods have shown varying effectiveness in optimizing LLMs. For example, LLM-Pruner (Ma et al., 2023a) employs structured pruning to selectively remove non-critical coupled structures based on gradient information, and it requires a fine-tuning step to recover the performance of the pruned models, which may take several hours. SliceGPT (Ashkboos et al., 2024) replaces each weight matrix with a smaller dense matrix, thereby reducing the embedding dimension of the network. However, this approach requires the application of principal component analysis for each linear layer, which can be time-consuming. Additionally, it introduces two auxiliary transformation matrices in each decoder block, which mitigates the overall effectiveness of the pruning. FLAP (An et al., 2024) formulates structured importance metrics and adaptively searches for a globally compressed model. Although it is relatively efficient, it often compromises performance. Shen et al. (2024) introduces a neural architecture search-based approach (denoted as NASLLM hereafter) to identify subnets of the original LLMs and implements a reformation algorithm to rectify the inherited weights. However, due to its searching nature, this method can be time-consuming.

Given the limitations of current structured pruning methods, particularly concerning time efficiency and the performance of pruned models, we propose a novel approach, FASP, that emphasizes both speed and accuracy. First, we introduce an innovative pruning structure that connects two sequential linear layers. This design enables the pruning of a subset of columns in the later layer, allowing for the corresponding rows in the preceding layer to be removed without adversely impacting performance. Inspired by Wanda (Sun et al., 2023), we develop a pruning metric for selecting columns to prune, which is both simple and efficient. Finally, we implement an effective restoration mechanism to recover the performance of the pruned model by leveraging inherited weights, thereby enhancing overall model fidelity.

We conduct comprehensive experiments on the OPT (Zhang et al., 2022) and LLaMA (Touvron et al., 2023) model families, comparing the proposed FASP against state-of-the-art techniques. Our results demonstrate that FASP achieves superior performance in terms of perplexity and accuracy on downstream tasks across these models. Specifically, we evaluate the pruning time for models of varying sizes, finding that FASP requires approximately 17 seconds to prune the OPT-125M model and about 20 minutes for the LLaMA-30B model on a single NVIDIA RTX 4090 GPU, highlighting substantial improvements in pruning speed. These findings emphasize the advantages of our method in practical applications, showcasing its potential to optimize LLMs efficiently while maintaining high performance.

## 2 BACKGROUND AND RELATED WORK

Pruning techniques in neural networks can be broadly categorized into unstructured and structured pruning. Unstructured and semi-structured pruning remove individual weights or partial substructures of a model to reduce its size. These methods, such as SparseGPT (Frantar & Alistarh, 2023), Wanda (Sun et al., 2023), FPSTAPruner (Zhao et al., 2024) and MaksLLM (Fang et al., 2024), often

achieve high sparsity levels without severely degrading model performance. However, they come with practical limitations. Unstructured pruning may provide limited computational speedup and memory savings due to its irregular sparsity structure and the necessity of storing indices. Additionally, the inference of semi-structured sparsified LLMs typically requires specialized hardware, such as NVIDIA's Ampere architecture GPUs (Mishra et al., 2021), which restricts their applicability in real-world scenarios.

In contrast, structured pruning entirely removes substructures from rows or columns in the weight matrices to attention heads, layers, or blocks. This results in direct memory savings and inference speedups that are achievable on any hardware, making it more practical for deployment. However, structured pruning imposes stricter constraints on what can be pruned, and the model's performance typically degrades more significantly. To compensate for this, retraining is traditionally required to restore the model's accuracy (Ma et al., 2023b). Retraining, though, is computationally expensive and time-consuming, particularly for LLMs. This makes pruning with retraining less feasible in practice, especially in scenarios requiring fast model adaptation and deployment. Given these challenges, there is a growing need for structured pruning methods that avoid retraining while maintaining model performance. Recent research has focused on developing such methods to improve the practicality of pruning large-scale models.

One line of work focuses on pruning larger components of LLMs. For instance, FinerCut (Zhang et al., 2024) prunes entire attention or feed-forward network blocks through a greedy algorithm that exhaustively evaluates and removes the blocks with minimal impact on the model's output. Similarly, ShortGPT (Men et al., 2024) removes entire decoder layers to achieve sparsity. Another direction focuses on pruning the weights of the linear layers in LLMs. SliceGPT (Ashkboos et al., 2024) introduces a rotation-based pruning method where the weights are first rotated by orthogonal matrices derived from principal component analyses (PCA) of the activations and the less important portions of the weights are pruned. However, this method introduces additional memory and computation overhead, as the orthogonal matrices must be stored to transform the residual connections during inference. Moreover, SliceGPT's pruning strategy solely relies on the activations and thus requires a large calibration dataset and high precision (64-bit) for PCA calculations, making it computationally expensive and memory-intensive. FLAP (An et al., 2024) proposes a novel pruning metric that measures the stability of the channels to determine which parts of the network to prune. A bias update mechanism is then used to compensate for the pruned parts. However, FLAP does not update the remaining weights after pruning, which contains much more parameters than the bias vector. Thus, incorporating bias-only compensation misses vast opportunities to remedy the performance loss due to pruning. NASLLM (Shen et al., 2024) explores network architecture search (NAS) methods for finding optimal subnets within LLMs and employs the alternating direction method of multipliers (ADMM) to determine the optimal updates for the remaining weights. However, these involve slow and inefficient search iterations, requiring approximately 5 hours to prune the LLaMA-7B model, as reported in the paper.

## 3 METHODOLOGY

In this section, we first provide a detailed overview of the proposed pruning structure. Next, we discuss the pruning metric inspired by Wanda. Finally, we introduce our efficient restoration mechanism for the pruned weights, aimed at enhancing overall model fidelity.

### 3.1 PRUNING STRUCTURE

Consider a 2-layer perceptron with weights $W_1 \in \mathbb{R}^{n \times p}$ and $W_2 \in \mathbb{R}^{m \times n}$ along with biases $b_1 \in \mathbb{R}^n$ and $b_2 \in \mathbb{R}^m$, given an input activation $X \in \mathbb{R}^{p \times q}$, the forward pass can be expressed as follows:

$$f(X) = W_2(\sigma(W_1 X + b_1)) + b_2, \tag{1}$$

where $\sigma(\cdot)$ is an element-wise activation function such as ReLU.

In this setup, each row of $W_1$ is directly associated with the corresponding column of $W_2$. Specifically, the $i$-th row of $W_1$ connects to the $i$-th column of $W_2$, reflecting the influence of the $i$-th hidden unit on the output:

$$f_i(X) = W_2[:, i] \cdot \sigma\left(W_1[i, :] X + b_1[i]\right) + b_2, \tag{2}$$

where $W_2[:, i] \in \mathbb{R}^{n \times 1}$ represents the $i$-th column of $W_2$ and $W_1[i, :] \in \mathbb{R}^{1 \times m}$ represents the $i$-th row of $W_1$. In structured pruning, pruning the $i$-th column of $W_2$ eliminates the contribution from the $i$-th hidden unit, resulting in the following output equation:

$$f_i(X) = \mathbf{0}^{m \times 1} \cdot \sigma\left(W_1[i, :]X + b_1[i]\right) + b_2 = b_2. \tag{3}$$

Consequently, we can also remove the $i$-th row of $W_1$ and $i$-th element of $b_1$. The output computation simplifies to:

$$f(X) = W_2'\left(\sigma(W_1'X + b_1')\right) + b_2, \tag{4}$$

where $W_2'$ represents $W_2$ with the $i$-th column removed, $W_1'$ denotes $W_1$ with the $i$-th row removed, and $b_1'$ signifies $b_1$ with the $i$-th element eliminated. Conversely, if we prune the $i$-th row of $W_1$, the $i$-th column of $W_2$ can also be removed directly.

Therefore, we can achieve efficient structured pruning by leveraging the inherent connections between the two sequential layers. This approach not only simplifies the model but also preserves performance, as it eliminates unnecessary computations without compromising the model's expressive capability.

As illustrated in Figure 1, the OPT architecture (Zhang et al., 2022) features decoder blocks that typically consist of self-attention mechanisms followed by feed-forward networks, which include two fully connected layers denoted by $W_{fc_1}$ and $W_{fc_2}$. Consequently, we can apply our pruning strategy by pruning the $i$-th column of $W_{fc_2}$ while simultaneously removing the $i$-th row of $W_{fc_1}$ and the $i$-th element of $b_{fc_1}$.

In LLaMA (Touvron et al., 2023), fully connected layers are also present within the *down*, *gate*, and *up* layers. We denote their weights as follows: $W_{\text{down}} \in \mathbb{R}^{d \times d_{\text{down}}}$, $W_{\text{gate}} \in \mathbb{R}^{d_{\text{down}} \times d}$, and $W_{\text{up}} \in \mathbb{R}^{d_{down} \times d}$. The forward pass through these layers can be described as:

$$\begin{cases} f_{up}(X) = W_{up}X, & \text{(5a)} \\ f_{gate}(X) = W_{gate}X, & \text{(5b)} \\ f_{down}(X) = W_{down}\left(f_{up}(X) \odot \sigma(f_{gate}(X))\right), & \text{(5c)} \end{cases}$$

where $\odot$ is the element-wise product. Therefore, when pruning the $i$-th column of $W_{down}$, we can simultaneously eliminate the $i$-th row of both $W_{up}$ and $W_{gate}$ without incurring additional performance loss, while achieving a gain in sparsity.

Next, we examine a self-attention block, which comprises four linear layers: $W_Q \in \mathbb{R}^{d \times d}$, $W_K \in \mathbb{R}^{d \times d}$, $W_V \in \mathbb{R}^{d \times d}$, and $W_O \in \mathbb{R}^{d \times d}$. The forward pass through these layers is described as follows:

$$\begin{cases} Q = W_QX, \quad K = W_KX, \quad V = W_VX, & \text{(6a)} \\ Attn = \text{softmax}\left(\dfrac{QK^T}{\sqrt{d}}\right)V, & \text{(6b)} \\ OUT = W_O(Attn). & \text{(6c)} \end{cases}$$

Therefore, since $W_O$ and $W_V$ interact, we can remove the $i$-th column of $W_O$ and subsequently eliminate the $i$-th row of $W_V$. Additionally, $W_Q$ and $W_K$ interact through their rows due to the term $QK^\top$, allowing us to remove the corresponding rows from both matrices.

However, our experiments (see Section 4.2) demonstrate that removing rows from $W_Q$ and $W_K$ significantly degrades the performance of the pruned model. Consequently, we opt not to prune these two layers; instead, we increase the sparsity level of the other layers uniformly to satisfy the overall sparsity requirements.

## 3.2 Pruning Metric

As shown in Figure 2, consider a linear layer with weights $W \in \mathbb{R}^{m \times n}$ and the corresponding input activations $X \in \mathbb{R}^{n \times p}$. For each individual weight, Wanda (Sun et al., 2023) evaluates its importance by calculating the product of its magnitude and the norm of the corresponding input feature. Specifically, the score for the weight $W_{i,j}$ is defined as follows:

$$S_{ij} = |W_{ij}| \cdot \left\|X_{(:, j)}\right\|_2, \tag{7}$$

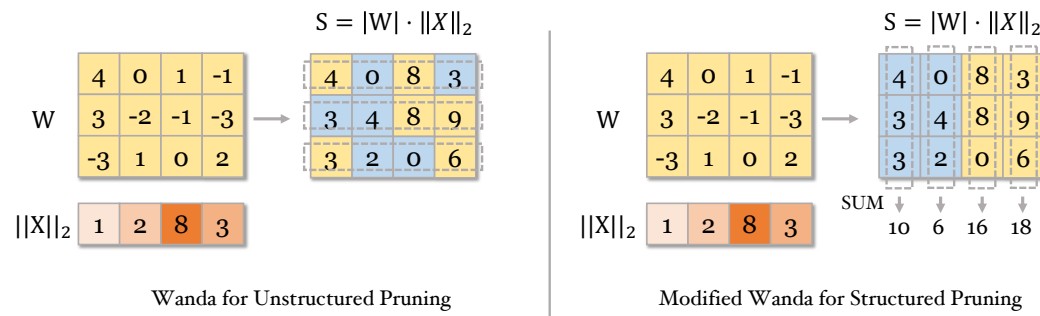

Figure 2: Illustration of the modified Wanda's metric for structured pruning, which removes the columns of $W$ where the corresponding columns in $S$ have smaller column-wise sums.

where $|\cdot|$ represents the absolute value operator and $\left\|X_{(:,j)}\right\|_2$ evaluates the $\ell_2$-norm of $j$-th column of $X$ The final score is computed by the product of these two scalar values.

Inspired by Equation 7, we extend this metric to structured pruning scenarios. Instead of comparing the magnitudes of individual elements in $S$ to determine which weights to prune, we compute the column-wise sums of $S$ and use these results for pruning decisions. Specifically, we remove the entire columns of $W$ where the corresponding columns in $S$ exhibit smaller column-wise sums.

The computational complexity of our metric primarily arises from the element-wise products and sums, yielding a complexity of $\mathcal{O}(mn)$, making it significantly efficient. In comparison, other importance scores guiding pruning decisions, such as SparseGPT (Frantar & Alistarh, 2023) and Pruner-Zero (Dong et al., 2024), have higher computational demands. SparseGPT requires an additional step to estimate the Hessian matrix, resulting in a complexity of $\mathcal{O}(mn^2 + n^3)$. Meanwhile, Pruner-Zero relies on gradient information to compute the importance score, necessitating a backward pass, which is computationally intensive.

In addition to its computational efficiency, we will demonstrate the pruning performance of this metric in the subsequent section on experimental results, highlighting its robustness and superiority. Together, these factors underscore the efficacy and efficiency of our proposed metric for structured pruning.

### 3.3 Restoration of the Pruned Weights

After identifying the columns to be deleted for the down and out projection operators, we address the problem of optimally updating the remaining weights to compensate for the pruning loss. Let $W^* \in \mathbb{R}^{m \times n}$ represent the pruned weight matrix of the down/out projection operator under sparsity $s$, with non-zero columns indexed by the set $M \in \{0, 1, \ldots, n-1\}^{n(1-s)}$. Additionally, let $W \in \mathbb{R}^{m \times n}$ be the dense weight matrix and $X \in \mathbb{R}^{n \times p}$ the activations from the preceding layer. The problem can be formulated as the following least-squares optimization:

$$\min_{W^*_{:,M} \in \mathbb{R}^{m \times (n \cdot (1-s))}} \frac{1}{2} \left\| W^*_{(:,M)} X_{(M,:)} - WX \right\|_F^2 .$$

Solving this optimal update problem is particularly challenging for unstructured sparsity, as it necessitates repeatedly solving linear systems for each row of $W^*$. However, due to the column-wise sparsity structure, the optimal solution $W^*_{(:,M)}$ can be efficiently obtained by computing the following normal equation once, where the term $\delta I$ with $\delta > 0$ is added to enhance numerical stability:

$$W^*_{(:,M)} = WXX_{(M,:)}^\top \left( X_{(M,:)} X_{(M,:)}^\top + \delta I \right)^{-1} . \tag{8}$$

It is noteworthy that NASLLM (Shen et al., 2024) proposes solving this problem via ADMM, which requires pre-computing the term $(XX^\top + \rho I)^{-1}$ for the ADMM iterations, where $\rho$ is the augmented Lagrangian penalty parameter. However, the computational complexity of obtaining this inverse is already equivalent to that of computing equation 8, not to mention the additional complexity of the

subsequent iteration steps. Furthermore, ADMM is an iterative algorithm that converges slowly as it approaches the optimal solution, inevitably facing a trade-off between efficiency and accuracy. In contrast, our proposed restoration method is both accurate and efficient.

## 4 EXPERIMENTS

In this section, we validate the performance and efficiency of FASP through comprehensive experiments. We begin by detailing our experimental settings, followed by a comparison of perplexity and zero-shot results for the pruned models obtained via FASP and various baseline methods, as well as the time taken for pruning.

**Models and baseline methods.** We compare FASP with state-of-the-art methods including SliceGPT (Ashkboos et al., 2024), NASLLM (Shen et al., 2024), FLAP (An et al., 2024), and LLM-Pruner (Ma et al., 2023b) on the LLaMA (Touvron et al., 2023) and OPT (Zhang et al., 2022) families with sizes ranging from 125m to 30B downloaded from HuggingFace's Transformers library (Wolf et al., 2019).

**Datasets and benchmarks.** We evaluate the perplexity of the models under various sparsity pruned by 128 randomly drawn calibration samples from the WikiText2 (Merity et al., 2016) dataset with the 2048 sequence length. Additionally, following the settings in (An et al., 2024; Ma et al., 2023b; Shen et al., 2024), we compare the zero-shot accuracy on standard common reasoning datasets including BoolQ (Clark et al., 2019), PIQA (Bisk et al., 2020), HellaSwag (Zellers et al., 2019), WinoGrande (Sakaguchi et al., 2021), OpenbookQA (Mihaylov et al., 2018), ARC-easy and ARC-challenge(Clark et al., 2018).

**Implementation details.** We implement FASP using the PyTorch framework (Paszke et al., 2019) and perform pruning on NVIDIA RTX 4090 GPUs, each equipped with 24GB of memory. To evaluate its performance, we utilize the official implementations of SliceGPT and FLAP. Since the code for NASLLM has not been released, we rely on the results reported in their paper for comparison. For LLM-Pruner, which necessitates a costly fine-tuning process to recover performance, we directly reference the results provided by the NASLLM authors. Our observations indicate that, on a single GPU, FASP can effectively prune 30B-level LLMs, while SliceGPT is limited to pruning models up to 13B due to the high memory requirements associated with PCA computations and the storage of additional orthogonal matrices.

### 4.1 MAIN RESULTS

**Perplexity and zero-shot results.** We present the perplexity results for the pruned OPT and LLaMA models of various sizes on WikiText in Tables 2 and 3, as well as in Figures 3 and 4. The data and figures consistently demonstrate that FASP outperforms LLM-Pruner, SliceGPT, NASLLM, and FLAP. We were unable to obtain the perplexity results for SliceGPT and NASLLM on the LLaMA-30B model, as SliceGPT requires more memory than the RTX 4090 GPU can provide for pruning, while the code for NASLLM is not available. Additionally, the zero-shot results for the pruned LLaMA-7B models are presented in Table 4, where FASP outperforms the best baseline method by 1.54% and 1.77% at 10% and 20% sparsity levels, respectively.

**Pruning time.** We compare the time required to prune LLaMA models among FASP and the baseline methods, as shown in Table 1. For NASLLM and LLM-Pruner, the only reported time is for pruning the LLaMA-7B model, which takes 5 hours and 3 hours on a single NVIDIA A100 GPU, respectively. The results for FLAP, SliceGPT, and FASP are obtained using NVIDIA RTX 4090 GPUs. Our findings show that FASP, which demonstrates the best pruning effectiveness, is several magnitudes faster than NASLLM and LLM-Pruner, approximately 10× faster than SliceGPT, and as efficient as FLAP in terms of pruning time.

**Inference speedup.** We evaluate the inference speedup of LLMs pruned by FASP on the NVIDIA RTX 4090. The OPT-2.7B model, pruned to 30% sparsity by FASP, achieves a 16% increase in end-to-end inference speed in our current implementation. We anticipate that this speedup can be further enhanced through more advanced programming techniques.

| Method | LLaMA-7B | LLaMA-13B | LLaMA-30B |
|---|---|---|---|
| NASLLM | 300min | - | - |
| LLM-Pruner | 180min | - | - |
| SliceGPT | 44min | 68min | 211min |
| FLAP | 5min | 8min | 22min |
| FASP (ours) | **4min** | **6min** | **20min** |

Table 1: Comparison of pruning times: FASP outperforms state-of-the-art methods.

| Method | Sparsity | OPT-125M | OPT-1.3B | OPT-2.7B |
|---|---|---|---|---|
| Dense | 0% | 27.66 | 14.63 | 12.47 |
| SliceGPT | 10% | 35.31 | 16.74 | 14.10 |
| NASLLM | 10% | 30.97 | 15.51 | 13.32 |
| FASP (ours) | 10% | **28.53** | **14.80** | **12.42** |
| SliceGPT | 20% | 54.88 | 20.17 | 16.81 |
| NASLLM | 20% | 44.12 | 19.23 | 16.44 |
| FASP (ours) | 20% | **30.55** | **15.64** | **14.19** |
| SliceGPT | 30% | 84.16 | 28.53 | 24.12 |
| NASLLM | 30% | 80.84 | 26.82 | 23.48 |
| FASP (ours) | 30% | **34.67** | **17.81** | **15.11** |

Table 2: WikiText perplexity (↓) of pruned OPT models under various sparsity. FASP drastically outperforms state-of-the-art methods.

## 4.2 ABLATION STUDIES

We assess the effectiveness of our pruning structure design through the ablation experiment presented in Table 5. In this experiment, we compare the results obtained from pruning all operators' columns using evenly distributed sparsity, with column selection performed by Wanda and optimal update, against those from the default setting of FASP. In the latter, the pruned rows and columns are correlated, and the $k$ and $q$ projections are skipped as described in Section 3.1. Our findings demonstrate that our pruning structure yields significantly better results.

Additionally, we evaluate the effectiveness of our strategy of not pruning the $W_Q$ and $W_K$ layers from the self-attention mechanism on OPT-125M, as detailed in Table 6. The row labeled *Pruning $W_Q$ and $W_K$* indicates the results obtained by pruning the rows of $W_Q$ and the corresponding rows of $W_K$, applying evenly distributed sparsity across layers. In contrast, the row labeled *FASP* reflects our default setting, where $W_Q$ and $W_K$ remain unpruned, and the sparsity is scaled to ensure consistent overall sparsity. Our observations reveal that pruning the rows of $W_K$ and their corresponding rows

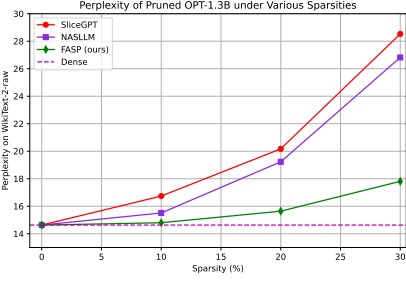

(a) Perplexity-vs-sparsity on OPT-1.3B.

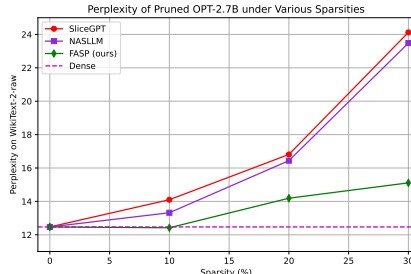

(b) Perplexity-vs-sparsity on OPT-2.7B.

Figure 3: Comparative analysis of sparsity versus perplexity across different methods for OPT-1.3B and OPT-2.7B models on WikiText dataset.

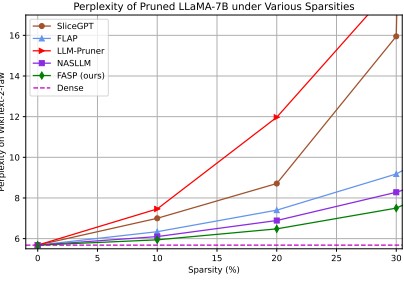

(a) Perplexity-vs-sparsity on LLaMA-7B.

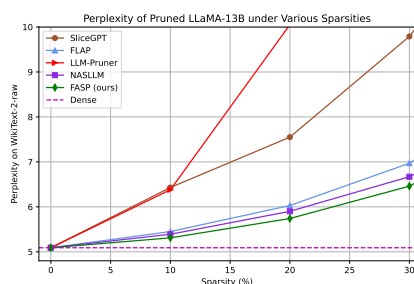

(b) Perplexity-vs-sparsity on LLaMA-13B.

Figure 4: Comparative analysis of sparsity versus perplexity across different methods for LLaMA-7B and LLaMA-13B models on WikiText dataset.

| Method | Sparsity | LLaMA-7B | LLaMA-13B | LLaMA-30B |
|---|---|---|---|---|
| Dense | 0% | 5.68 | 5.09 | 4.10 |
| LLM-Pruner | 10% | 7.46 | 6.38 | - |
| SliceGPT | 10% | 7.00 | 6.43 | - |
| NASLLM | 10% | 6.10 | 5.39 | **4.44** |
| FLAP | 10% | 6.34 | 5.45 | 4.52 |
| FASP (ours) | 10% | **5.94** | **5.31** | 4.46 |
| LLM-Pruner | 20% | 11.97 | 10.05 | - |
| SliceGPT | 20% | 8.71 | 7.55 | - |
| NASLLM | 20% | 6.89 | 5.90 | 4.94 |
| FLAP | 20% | 7.40 | 6.03 | 5.18 |
| FASP (ours) | 20% | **6.48** | **5.74** | **4.93** |
| LLM-Pruner | 30% | 18.58 | 22.36 | - |
| SliceGPT | 30% | 15.95 | 9.79 | - |
| NASLLM | 30% | 8.28 | 6.67 | 5.63 |
| FLAP | 30% | 9.18 | 6.97 | 6.28 |
| FASP (ours) | 30% | **7.50** | **6.46** | **5.51** |

Table 3: WikiText perplexity ($\downarrow$) of pruned LLaMA models under various sparsity. FASP outperforms state-of-the-art methods.

| Method | Sparsity | BoolQ | PIQA | HellaSwag | WinoGrande | ARC-e | ARC-c | OBQA | Mean |
|---|---|---|---|---|---|---|---|---|---|
| Dense | 0% | 75.11 | 79.16 | 76.22 | 70.09 | 72.94 | 44.71 | 44.40 | 66.09 |
| LLM-Pruner | 10% | 67.95 | 77.42 | 69.31 | 63.54 | 66.33 | 39.85 | **41.20** | 60.80 |
| SliceGPT | 10% | 57.68 | 69.80 | 59.32 | 68.11 | 62.75 | 36.01 | 38.00 | 55.95 |
| FLAP | 10% | **74.43** | 75.41 | 68.68 | 67.01 | 65.78 | 38.48 | 41.00 | 61.54 |
| NASLLM | 10% | 74.37 | 76.88 | 70.71 | 67.56 | 68.39 | 40.10 | 39.20 | 62.46 |
| FASP (ours) | 10% | 73.15 | **77.53** | **74.11** | **68.90** | **70.45** | 42.92 | 41.00 | **64.00** |
| LLM-Pruner | 20% | 59.39 | 75.57 | 65.34 | 61.33 | 59.18 | 37.12 | 39.80 | 56.82 |
| SliceGPT | 20% | 37.89 | 64.09 | 45.67 | 62.75 | 53.62 | 31.74 | 33.20 | 46.99 |
| FLAP | 20% | 68.59 | 74.21 | 64.98 | 64.40 | 50.89 | 37.80 | 40.20 | 58.58 |
| NASLLM | 20% | **70.98** | 74.92 | 67.29 | 64.64 | 64.23 | 36.52 | 39.40 | 59.71 |
| FASP (ours) | 20% | 69.36 | **75.95** | **69.40** | **66.77** | **68.27** | **40.19** | 40.40 | **61.48** |

Table 4: Zero-shot results (accuracy, $\uparrow$) of the pruned LLaMA-7B models under various sparsity. FASP outperforms state-of-the-art methods.

of $W_Q$ significantly diminishes the performance of the pruned model, thereby underscoring the effectiveness of our pruning strategy.

|         | 10%   | 20%   | 30%   |
|---------|-------|-------|-------|
| Wanda   | 30.61 | 39.28 | 54.89 |
| FASP    | **28.53** | **30.55** | **34.67** |

Table 5: WikiText perplexity ($\downarrow$) of pruned LLaMA models under various sparsity.

|                        | 10%   | 20%   | 30%   |
|------------------------|-------|-------|-------|
| Pruning $W_Q$ and $W_K$ | 44.13 | 61.62 | 83.65 |
| FASP                   | **28.53** | **30.55** | **34.67** |

Table 6: WikiText perplexity ($\downarrow$) of pruned LLaMA models under various sparsity.

## 5 DISCUSSION

In this section, we reflect on the limitations of FASP. Despite its strengths, one limitation is the decision to forgo pruning the rows of $W_Q$ and $W_K$ in the self-attention layers. While our experiments demonstrated that pruning these layers degrades performance, future work could explore more sophisticated strategies for dealing with these components, such as adaptive sparsity levels or selective pruning criteria that may mitigate the performance loss.

## 6 CONCLUSION

In this paper, we present FASP, a fast and accurate structured pruning method designed for LLMs. By capitalizing on the inherent connections between sequential layers, FASP allows for efficient pruning with minimal performance drop. Our proposed pruning structure and metric streamline the pruning process, while the restoration mechanism effectively recovers model fidelity. Through comprehensive experiments on the OPT and LLaMA model families, we demonstrate that FASP significantly outperforms existing pruning techniques both in terms of speed and accuracy. Notably, FASP is capable of pruning large-scale models like LLaMA-30B in a fraction of the time required by other methods, while maintaining competitive performance on perplexity and zero-shot evaluation tasks. These results highlight the potential of FASP to improve the deployment efficiency of LLMs across diverse hardware platforms, including those with limited computational resources.

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
