# OpenReview forum: "FASP: Fast and Accurate Structured Pruning of Large Language Models"
_ICLR.cc/2025/Conference — ICLR 2025 Conference Withdrawn Submission_

### Official Review · Reviewer_czA3 · 2024-10-25

**Soundness:** 2
**Presentation:** 1
**Contribution:** 1
**Rating:** 3
**Confidence:** 5

**Summary:**

This paper proposes FASP (Fast and Accurate Structured Pruning), a structured pruning algorithm for LLMs that emphasizes both pruning efficiency (speed) and accuracy.
The authors' main ideas are (1) formulation for pruning neurons, (2) Wanda[1]-based importance metric, and (3) restoration method after pruning.
The authors demonstrate the effectiveness of FASP by showing that FASP shows the highest accuracy (or lowest perplexity) and the fastest pruning speed among competitors.

[1] Sun, Mingjie, et al. "A simple and effective pruning approach for large language models." arXiv preprint arXiv:2306.11695 (2023).

**Strengths:**

FASP, the proposed method, is simple and easy to understand. Despite its simplicity, FASP shows nice performance; FASP achieves the highest accuracy (or the lowest perplexity) in almost all settings while requiring the shortest time for pruning.
Combined with the fact that the small model pruned with FASP shows a 16% inference speedup, this simple method can be considered being useful in practical settings.

**Weaknesses:**

### Novelty
1. The main weakness of this paper lies in its novelty. This paper proposes (1) formulation, (2) importance metric, and (3) restoration method, but all of these ideas can be found in previous works [1, 2, 3, 4] with slight modifications in some cases.
In detail, the pruning of neurons is used in [1,2,3] and the importance metric of FASP is a straightforward modification of Wanda [4]; it just sums up the importance score of weights in each column to measure the importance of the column. The restoration method of FASP which solves the least-square problem is also used in previous works [1,2].
Therefore, FASP has limited novelty and originality.

### Experiments
2. First of all, what is the definition of the problem that FASP tries to solve? Is the compression constraint related to latency or model size? According to the definition of your problem, you need to add more competitors to verify the performance of FASP; you need to compare FASP with layer pruning algorithms, e.g., SLEB [5] if your problem includes latency constraints or 2:4 pruning algorithms, e.g. Wanda [4], if your problem includes model size in your problem. In other words, there are lack of justifications for selecting baseline methods.

3. The inference speed of the pruned model is quite important since the acceleration after pruning is the major reason why we use structured pruning rather than unstructured pruning. However, there are lack of experiments about the inference speed of the pruned models. The only experimental results can be found in four lines in Section 4.1, but there is no comparison with baseline methods. There is no detailed explanation of the experimental settings, e.g., the lengths of input and output sequences.

4. The authors use old-fashioned models such as OPT and Llama. Considering that Llama3 8B shows more than 30%p higher accuracy than Llama2 7B on ARC-challenge benchmark, recent models might show different patterns to the old-fashioned models used in this paper.

### Writing
5. Figures and tables are not arranged properly. For example, Tables 5 and 6 can be placed in a single line. Tables 1 and 2 are also can be placed in a single line if you control column widths.

6. Font sizes in Figures 3 and 4 are too small and hard to read.

7. According to the guidelines of ICLR, table captions must be located on the upper side of the tables.

8. There are lack of explanation about Figure 2 in its caption. What's the meaning of each color?

9. The authors use Section 5 for discussion, but there are only 5 lines of text in this section and lack of meaningful content.

10. The authors use imprecise mathematical notations, such as n(1-s) in line 254; there is no guarantee that n(1-s) is an integer and it requires a floor or ceiling function to become an integer.

11. The name of the proposed method does not contain any characteristics of the proposed method and it is too general. According to the reported experimental results, the method is not significantly "fast" and not significantly "accurate"

12. There is no reproducibility statement which is encouraged by the conference guideline (https://iclr.cc/Conferences/2025/AuthorGuide).

[1] Park, Seungcheol, Hojun Choi, and U. Kang. "Accurate Retraining-free Pruning for Pretrained Encoder-based Language Models." The Twelfth International Conference on Learning Representations. 2024.

[2] Kwon, Woosuk, et al. "A fast post-training pruning framework for transformers." Advances in Neural Information Processing Systems 35 (2022): 24101-24116.

[3] Ma, Xinyin, Gongfan Fang, and Xinchao Wang. "Llm-pruner: On the structural pruning of large language models." Advances in neural information processing systems 36 (2023): 21702-21720.

[4] Sun, Mingjie, et al. "A simple and effective pruning approach for large language models." arXiv preprint arXiv:2306.11695 (2023).

[5] Song, Jiwon, et al. "SLEB: Streamlining LLMs through Redundancy Verification and Elimination of Transformer Blocks." arXiv preprint arXiv:2402.09025 (2024).

**Questions:**

1. Could you explain the novelty of this paper if I missed some points?

2. Is there any reason for using OPT and Llama model families?

3. Could you explain some intuitions that explain the phenomena that $W_Q$ and $W_K$ are hard to prune? In my opinion, the answer to this question would improve the quality of Section 5.

4. Is there any reason for pruning neurons in MHA sublayers rather than attention heads as in previous works [1, 2]?

[1] Park, Seungcheol, Hojun Choi, and U. Kang. "Accurate Retraining-free Pruning for Pretrained Encoder-based Language Models." The Twelfth International Conference on Learning Representations. 2024.

[2] Kwon, Woosuk, et al. "A fast post-training pruning framework for transformers." Advances in Neural Information Processing Systems 35 (2022): 24101-24116.

---

### Official Review · Reviewer_yczM · 2024-10-31

**Soundness:** 2
**Presentation:** 2
**Contribution:** 2
**Rating:** 5
**Confidence:** 4

**Summary:**

Considering both speed and accuracy of structured pruning for LLMs, this paper propose a novel structured pruning framework named FSAP, which achieves efficient structured pruning by leveraging the inherent connections between sequential layers and the restoration of pruned weights. Experiments on the OPT and LLaMA model families demonstrates the effectiveness of FSAP in terms of perplexity and accuracy on downstream tasks.

**Strengths:**

1. The paper is generally well-written, with experiments across different types of models and tasks, showing interesting results.
2. Leveraging the inherent position mapping in matrix multiplication to reduce substructures from rows and columns in the weight matrices is a good idea.

**Weaknesses:**

1. The proposed pruning structure does not seem to be universal and needs to be specifically designed for different models (such as OPT and LLaMA mentioned in this paper).
2. Figures 1 and 2 are oversimplified to summarize the characteristics of the proposed method.
3. In terms of Tables 1,3 and 4, the proposed FSAP has only a slight improvement in pruning time and accuracy compared to FLAP.
4. It is inappropriate to use the same title for Tables 3, 5, and 6, which have different purposes.
5. Only comparative and ablation experiments were covered, lacking some intuitive analytical experiments to explore the intrinsic mechanisms of the proposed FSAP.
6. More experiments or explanations are needed to analyze why the proposed FSAP should not remove the columns of WQ and WK.
7. The authors did not provide any code for review, so the reproducibility of this paper is open to question.
8. The prune metric is similar to Wanda.

**Questions:**

1. How effective is the proposed method on other models? Such as Vicuna, Mixtral, and Qianwen?

---

### Official Review · Reviewer_tqmK · 2024-11-01

**Soundness:** 3
**Presentation:** 2
**Contribution:** 2
**Rating:** 3
**Confidence:** 4

**Summary:**

In this paper, the authors propose a new structural pruning technique, FASP, that achives better performance and is also faster than privious baselines.

Specifically, the authors:
1. notice that the pruning of some columns in a weight matrix $W_2$ is associated with the corresponding rows in the precedding weight $W_1$. So one only needs to determine the pruned columns of $W_2$. The architectures of Transformer, like OPT and Llama series, also facilitates this setup.
2. determine the pruned columns based on a modified version of Wanda score. Originally, Wanda score only supports element-wise (i.e. unstructural) pruning. The authors add one more operation to Wanda, i.e. summing in the column dimension, and prune the column with the smallest sum. In this way, the Wanda score is adapted to structural pruning.
3. finally, restore the pruned weights based on least-squares optimization, i.e. ensuring the output of the pruned layer similar to the output of the original layer.

The authors test the pruned LLM with FASP on some benchmarks, i.e. WikiText and 7 commonsense reasoning tasks, and compare the performance with some well-known structural pruning methods with various sparsities. The experiments show that FASP is better than the baselines in a sparsity range of 10%-30%. In addition, the authors also find that keeping Q and K untouched is a better option, and show that FASP's inference is fatser than the original model by 16%.

**Strengths:**

1. The paper is well written and well motivated, I can fully understand the technique details.
2. The column-row corresponding pruning is clever.
3. The experiments are extensive, with enough support for the main claims.

**Weaknesses:**

1. **More perplexity results, like on C4.** The comparison of perplexity on WikiText might not be fair. FASP restores the pruned weights on 128 samples from WikiText, making it adapt to WikiText. It's an unfair comparison for methods that don't have this restoration step. I also observe a slightly overfit in Table 2, where FASP's with 10% sparsity can achives a smaller perplexity than the original LLM, i.e. 12.42 vs 12.47. Therefore, I encourage to include more perplexity results, like on C4 (but also use WikiText samples to restore the pruned weights).

2. **Lack of recent LLM series.** The LLMs used in this paper is OPT and LLaMA (first version), it's better to include some recent series, like Llama-3. It's known that Llama-3 is difficult for quantization, it might also pose challenge to pruning. I understand that most previous works report results on LLaMA and OPT, so you prefer to do experiments on them. But the results on a recent series is also important. You can only compare FASP to the strongest baseline on the recent series.

3. **Lack of novelty.** From the summary, The innovation from this work seems limited. The adapted Wanda is very similar to Wanda, only adding one summation operation along the column.  And the knowledge restoration is basically layer-wise knowledge distillation.

**Questions:**

1. According to L251, the down and out projection layers are used to determined the pruned columns. I think there is a missing ablation study. We can also determine the pruned rows of gate/up and V projection layer, and prune the corresponding columns. In a word, what is the reason to apply adapted Wanda score to columns instead of rows.

2. In this paper, the inference speed is an advantage of structural pruning over unstructural pruning. According to L322, the speedup of FASP is 16% for 30% sparsity, which seems not a lot. Can you report the speedup of semi-structural pruning on the same level of sparsity, like 2:4 or 4:8 and 30% sparsity with original Wanda score?

---

### Official Review · Reviewer_iYLx · 2024-11-04

**Soundness:** 3
**Presentation:** 3
**Contribution:** 3
**Rating:** 6
**Confidence:** 4

**Summary:**

This paper introduces FASP (Fast and Accurate Structured Pruning), a framework for efficiently pruning large language models (LLMs) with an emphasis on reducing both computational and memory demands without sacrificing model accuracy. Structured pruning methods like FASP aim to address the increasing resource constraints involved in deploying large-scale models by pruning layer-wise connections, thus directly impacting speed and efficiency. FASP introduces a novel pruning structure for paired layers and a restoration mechanism that recalibrates remaining weights to maintain model fidelity.

**Strengths:**

1. Effective Weight Restoration Mechanism
To counteract potential accuracy drops from pruning, FASP introduces a restoration mechanism that optimizes the remaining weights, preserving model fidelity. This component reinforces the pruning approach, ensuring that accuracy and perplexity remain competitive across various levels of sparsity, particularly valuable in maintaining generalization for downstream tasks.

2. Relevance to Real-World Applications
With its focus on deployment practicality, FASP is particularly relevant for resource-constrained environments, including edge devices and mobile deployments. The method’s combination of pruning speed, accuracy retention, and structural adaptability makes it a compelling choice for applications where model size and computational overhead are primary concerns.

3. Clear and Comprehensive Experimentation
The experimental setup is well-executed, covering both perplexity and zero-shot task accuracy benchmarks on a range of models, from OPT-125M to LLaMA-30B.

4. FASP’s pruning approach is carefully calibrated to achieve high sparsity without the need for retraining, unlike many other pruning methods that require extensive fine-tuning to regain performance. This balance is achieved through a design that leverages the inherent connections between sequential layers and an efficient restoration mechanism. By avoiding retraining, FASP reduces the resource and time investment typically required post-pruning, making it highly practical for settings where quick, resource-light model compression is essential.

**Weaknesses:**

1. Lack of Memory Savings Analysis
The paper primarily reports on pruning speed and latency improvements, omitting a detailed analysis of memory savings. Given that structured pruning is often motivated by reductions in memory footprint, a quantitative comparison with other pruning methods on memory consumption would strengthen the paper’s practical impact claims. Including this metric would provide additional insight into FASP’s viability for memory-constrained deployments.

2. Missed Opportunity in Attention Mechanism Pruning
The approach skips pruning for 𝑊𝑄  and 𝑊𝐾 layers in self-attention due to observed performance degradation. While pragmatic, this choice leaves redundancy in these attention components. Exploring alternatives like partial pruning of heads or adaptive sparsity allocation within attention could increase FASP’s efficiency while addressing the performance concerns, allowing for a more comprehensive pruning strategy.

3. Limited Hardware Evaluation Scope
Benchmarking on edge devices and CPU inference frameworks should have been focused.
For a pruning method like FASP, which aims to make large models more accessible on constrained hardware, a broader evaluation on diverse hardware setups. Alternatively, a discussion on anticipated hardware bottlenecks or expected variations in performance would help users assess FASP’s versatility across deployment scenarios.

**Questions:**

1. On the Decision Not to Prune WQ and W_K Layers:  In Section 3.1, the authors mention that """our experiments... demonstrate that removing rows from WQ and WK significantly degrades the performance of the pruned model.""" Given this observation, have the authors considered more selective pruning strategies within these layers ? It would be helpful to understand if alternative approaches were explored to optimize these components without incurring the same level of performance loss.

2. Memory Savings Analysis While the paper highlights improvements in pruning speed and inference latency, there’s no mention of memory savings achieved post-pruning. Could the authors provide any insights into the actual memory reduction obtained through FASP compared to baseline methods? Given that memory efficiency is often a key motivation behind pruning, this additional data would help assess FASP’s overall impact on resource usage.  In fact, for inference on low-resource or edge devices, a memory analysis would be essential.

3. Edge Device and CPU Performance: Have the authors evaluated FASP’s performance on edge devices or CPUs? Since pruning is often intended to make models compatible with low-resource hardware, it would be beneficial to see results for these setups. Have the authors tried running and comparing performance using frameworks like Ollama, Onnx, llama.cpp, which are optimized for such environments? Plotting metrics from these tests could provide a clearer picture of FASP’s practical utility in real-world deployment scenarios.

---

### Official Review · Reviewer_kYwz · 2024-11-04

**Soundness:** 2
**Presentation:** 3
**Contribution:** 1
**Rating:** 3
**Confidence:** 5

**Summary:**

This paper introduces FASP, a structured pruning framework designed to enhance the efficiency and maintain the performance of large language models (LLMs) through structured pruning. FASP proposes a pruning strategy by connecting sequential layers, allowing it to remove redundant elements without retraining. It employs a modified pruning metric inspired by Wanda, combined with a restoration mechanism to compensate for pruning losses. The authors conduct experiments on OPT and LLaMA models, claiming that FASP achieves better perplexity and accuracy while reducing pruning times compared to other methods.

**Strengths:**

- The paper is clearly written, with well-organized sections detailing the methodology, experiment setup, and results.
- FASP aims at achieving efficiency on NVIDIA RTX 4090, making the method more accessible for practical applications.

**Weaknesses:**

- While FASP offers practical improvements, its core ideas rely heavily on existing pruning strategies, such as those proposed by Wanda and similar structured pruning frameworks. The novelty primarily lies in the integration of these techniques, which does not constitute sufficient methodological novelty.
- Some recent related work is missing. For example, DISP-LLM: Dimension-Independent Structural Pruning for Large Language Models.
- The experiment did not include the comparison with many existing work, such as the above DISP-LLM paper, Pruner-Zero, etc.

**Questions:**

Please refer to the weakness section.

---

### Author Response · Authors · 2024-12-04

We would like to sincerely thank the reviewers for their thoughtful and constructive feedback on our submission. While our proposed method, FASP, demonstrates clear practical advantages by surpassing state-of-the-art baselines in both pruning speed and performance, we acknowledge that the methodological innovation is primarily centered around the row-column corresponding pruning framework. Moving forward, we plan to build on this foundation by exploring more effective strategies for determining the pruned rows and columns, beyond relying on metrics inspired by Wanda. Additionally, we aim to investigate the observed performance drop when pruning the weights of K and Q projections as well as enriching the ablation studies to provide a more comprehensive understanding of these effects.

We are grateful for the opportunity to improve our work through the insightful comments provided, and we will incorporate this feedback as we refine our method for a future submission. Thank you again for your time and effort in reviewing our paper.

---

### Note · Authors · 2024-12-04

I have read and agree with the venue's withdrawal policy on behalf of myself and my co-authors.